

# Methylene blue inhibits nucleation and elongation of SOD1 amyloid fibrils

Greta Musteikyte[1,2], Mantas Ziaunys[2] and Vytautas Smirnovas[2]

[1] Department of Chemistry, University of Cambridge, Cambridge, United Kingdom
[2] Life Sciences Center, Institute of Biotechnology, Vilnius University, Vilnius, Lithuania

## ABSTRACT

Protein aggregation into highly-structured amyloid fibrils is linked to several neurodegenerative diseases. Such fibril formation by superoxide dismutase I (SOD1) is considered to be related to amyotrophic lateral sclerosis, a late-onset and fatal disorder. Despite much effort and the discovery of numerous anti-amyloid compounds, no effective cure or treatment is currently available. Methylene blue (MB), a phenothiazine dye, has been shown to modulate the aggregation of multiple amyloidogenic proteins. In this work we show its ability to inhibit both the spontaneous amyloid aggregation of SOD1 as well as elongation of preformed fibrils.

## INTRODUCTION

Lou Gehrig's disease or ALS (amyotrophic lateral sclerosis) is a late-onset fatal neurodegenerative disorder that results in upper motor neuron death and leads to gradual paralysis of the whole body. ALS is diagnosed for patients averagely at the age of 55, lasts for 3–5 years and eventually leads to death (*Taylor, Brown & Cleveland, 2016*). ALS occurs in both familial and sporadic form. The inherited form (fALS) accounts for 5–10% cases and is associated with mutations in more than 50 genes (*Taylor, Brown & Cleveland, 2016*; *Smith et al., 2017*), while the sporadic form (sALS) is diagnosed in all other cases (*Peters & Brown Jr, 2015*). The incidence of the disease is 1.9 cases of 100,000 per year and it is estimated that in 2040 the number of individuals suffering from the disorder will reach 376,674 (*Arthur et al., 2016*). Currently, there are no effective therapies for ALS; with only Riluzole and Edaravone being FDA approved drugs. However, they only prolong patient survival without curing the disease completely (*Miller et al., 2002*; *Rothstein, 2017*; *Fang et al., 2018*; *Dash, Babu & Srinivas, 2018*; *Jaiswal, 2019*).

SOD1 (superoxide dismutase I) is a 32-kDa homodimer which plays an antioxidative role in the cell. *SOD1* was the first gene to be associated with ALS in 1993 (*Rosen et al., 1993*) and more than 180 disease-related mutations have been reported so far (*Pansarasa et al., 2018*). It is still considered as one of the most commonly associated genes with the disorder (*Renton, Chiò & Traynor, 2014*). A significant number of studies revealed ALS association with SOD1 misfolding and aggregation (*Rakhit & Chakrabartty, 2006*; *Rotunno & Bosco, 2013*; *Lee & Kim, 2015*; *Pickles et al., 2016*; *Pansarasa et al., 2018*; *Paré et al., 2018*).

Corresponding author
Vytautas Smirnovas,
vytautas@smirnovas.info

According to the gain-of-function hypothesis, misfolded SOD1 forms cross- β-sheet rich amyloid deposits in the central nervous system (*Peters & Brown Jr, 2015*). Such deposits impede clearance of neurotransmitter glutamate after nerve signal transduction (*Boillée, Velde & Cleveland, 2006*; *Zarei et al., 2015*), inhibit oxidative phosphorylation (*Mattiazzi et al., 2002*), disrupt axonal mitochondrial transport (*Maniecka & Polymenidou, 2015*) and eventually lead to motor neuron death.

Different strategies are employed in drug discovery in the field of neurosciences. One of them is direct targeting of a disease-associated protein. This is realized by looking for agents that stabilize the native structure of the protein (*Maurer et al., 2018*) or screening for aggregation inhibitors and fibrillization pathway modulators. In the context of ALS, SOD1 is considered to be an extremely important therapeutic target to treat this particular disease (*Liu et al., 2012*; *Pansarasa et al., 2018*). A number of studies demonstrate SOD1 protein stabilizing and/or aggregation hampering effects caused by both synthetic and natural compounds. These include curcumin (*Bhatia et al., 2015*), flavonoids (*Zhuang et al., 2016*), epigallocatechin-3-gallate (*Srinivasan & Rajasekaran, 2017*), cisplatin (*Banci et al., 2012*), azauracil and uracil derivatives (*Nowak et al., 2010*), isoproterenol, 5-fluorouridine (*Wright et al., 2013*) and libraries of other compounds (*Ray et al., 2005*; *Anzai et al., 2016*).

Emerging potential of a small-molecule compound methylene blue (MB) as a therapeutic agent in the field of neurodegenerative disorders has also been discovered recently. MB is a phenothiazine dye used in medicine for various purposes ranging from nervous tissue visualization during surgery to methemoglobinemia, ifosfamide-induced toxicity, malaria and vasoplegic syndrome treatment (*Oz et al., 2011*). Its pharmacokinetic properties are well known; the typical oral dose in clinics ranges between 50–300 mg. The drug crosses the blood–brain barrier and is reported to interact with nitric oxide synthase, butyrylcholinesterase, acetylcholinesterase; it is also an inhibitor of voltage-gated calcium channels, adrenergic, NMDA, histamine H1/H2 and serotonin receptors (*Yamashita et al., 2009*; *Oz, Lorke & Petroianu, 2009*; *Oz et al., 2011*).

A range of studies report an inhibitory effect of MB on aggregation of various neurodegenerative disease-related proteins. It has been shown that MB reduces the amount of TDP-43 aggregates in the SH-SY5Y cell line (*Yamashita et al., 2009*) and recovers motor neuron function in *Danio rerio* and *Caenorhabditis elegans* ALS disease models (*Vaccaro et al., 2012*; *Vaccaro et al., 2013*). It has also been demonstrated that the compound inhibits huntingtin aggregation in vitro, increases the survival rate of mutant huntingtin transduced primary neuron culture and shows a positive effect in Huntington's disease models *in vivo* (*Sontag et al., 2012*). MB also inhibits Alzheimer's and Parkinson's disease-related tau aggregation and plaque formation (*Wischik et al., 1996*; *Akoury et al., 2013*; *Crowe et al., 2013*). The compound has been shown to impede Alzheimer's disease-associated amyloid β (Aβ) oligomerization as well. MB modulates the Aβ aggregation pathway by facilitating fibrillization of the peptide, therefore reducing the formation of highly toxic oligomer species (*Necula et al., 2007*). The effect of MB was also demonstrated on both the native structure, as well as aggregation of a model amyloidogenic protein—lysozyme (*Saha et al., 2018*). Three studies report on the effect of MB on SOD1 G93A mice (*Lougheed & Turnbull, 2011*; *Audet, Soucy & Julien, 2012*; *Dibaj et al., 2012*). However, this particular

mutation accounts for only a minor proportion of fALS cases (*Renton, Chiò & Traynor, 2014*). To our knowledge, there are no studies reporting about MB effect on wild-type SOD1 (wtSOD1) aggregation. This subject is of particular importance since most ALS cases (90–95%) are associated with wild-type proteins (*Renton, Chiò & Traynor, 2014*; *Peters & Brown Jr, 2015*).

In this study, we demonstrate for the first time that MB facilitates wtSOD1 "off-pathway" aggregation in vitro under destabilizing, amyloid-formation favouring conditions and give insights into the underlying molecular mechanism of this particular phenomenon.

## MATERIAL AND METHODS

### Stock solution preparation

Methylene blue (Sigma-Aldrich) was dissolved in deionized water in an approximate concentration of 10 mM and stored in the dark. The exact concentration was measured using a diluted MB solution with a UV-1800 Shimadzu spectrophotometer in a one cm pathlength cuvette ($\varepsilon_{664}$= 74,028 M$^{-1}$ cm$^{-1}$). The stock solution was filtered through a 0.2 $\mu$m syringe filter (Kinesis) before use.

### SOD1 synthesis and purification

Recombinant C-terminally his-tagged SOD1, cloned in pET303 vector, was expressed in *E. coli* BL21 Star (DE3) strain (Invitrogen). Culture from stock was grown overnight in ZYM-5052 autoinduction media (*Studier, 2005*) at +37 °C. Cells were harvested by centrifugation for 30 min at 6 000 rpm (HeroLab), 4 °C. The culture was first homogenized mechanically and then sonicated (VS70/T probe, Bandelin) in 50 mM sodium phosphate, 100 mM NaCl pH 7.5 buffer. Cell debris was pelleted by centrifugation for 30 min at 18 000 rpm, 4 °C. The supernatant was mixed with pre-equilibrated, Ni$^{2+}$ ion loaded IMAC resin (GE Healthcare) and left at 4 °C to equilibrate. The resin was loaded into GE Healthcare HiScale 26/40 column, the column was connected to *ÄKTApurifier* chromatographic system and target protein eluted with 200 mM imidazole step gradient. ApoSOD1 was obtained as described (*Chattopadhyay et al., 2008*) with slight modifications. Briefly, SOD1 was dialysed in 100 mM acetate, 50 mM NaCl, 10 mM EDTA pH 3.8 buffer overnight; then in 100 mM acetate, 50 mM NaCl, 50 mM EDTA pH 3.8 buffer for 4 h and two times in 10 mM potassium phosphate pH 7.4 buffer. Dialysed protein was filtered through a 0.22 $\mu$m filter (Millipore) and stored at −80 °C.

### Development of SOD1 aggregation assay

Wild-type metallated and disulfide-oxidized superoxide dismutase I is known to be an extremely stable enzyme with a melting temperature of 92 °C (*Culik et al., 2018*) and retains its activity in high concentrations of denaturing agents (*Arnesano et al., 2004*). Therefore, it is difficult to follow aggregation of wtSOD1 in vitro under physiological conditions in real-time. What is more, fibrillization kinetics of SOD1 are difficult to reproduce due to competing aggregation pathways and process stochasticity (*Abdolvahabi et al., 2016*). In this study, we have developed an assay allowing one to follow rapid and reproducible SOD1 kinetics at physiological pH (see 'Methods'). One key component used in the assay is a

disulfide reducing agent, since disulfide reduction in SOD1 leads to destabilization of the SOD1 dimer and facilitates fibrillization (*Furukawa, Torres & O'Halloran, 2004*; *Sheng et al., 2013*; *Culik et al., 2018*). Another key component is guanidine hydrochloride (GuHCl), a chaotropic agent that accelerates SOD1 aggregation (*Sheng et al., 2013*) and in our case, strongly promotes SOD1 fibrillization and prevents "off-pathway" oligomer formation (Fig. S1). In the absence of GuHCl, SOD1 forms round-shaped oligomers (Fig. S1A), whereas in the presence of 0.5 M GuHCl, thread-like amyloid fibers are formed (Fig. S1B). This most likely occurs due to chaotropic agent facilitated SOD1 unfolding and obtainment of a conformation that is prone to aggregation into intermolecular β-sheet rich amyloid fibers. In other words, GuHCl promotes the fibrillization pathway, but not the "off-pathway" oligomer formation. In our study, demetallated and disulfide bond reduced SOD1 is used, since demetallation, as well as disulfide bond reduction, leads to protein destabilization and an increased propensity to aggregate (*Furukawa & O'Halloran, 2005*; *Sheng et al., 2013*). To confirm that purified SOD1 is in the apo form, a thermal unfolding assay was performed (see Supplementary Methods section). The measured melting temperature of SOD1 is 50.3 ± 0.1 °C and corresponds to a disulfide-oxidized and demetallated form of the enzyme; this is in agreement with the data presented in literature (Fig. S2, *Furukawa & O'Halloran, 2005*). These SOD1 destabilizing conditions do not reflect the biological environment that the protein is found in vivo (apart from the pH value), however, theyare necessary to obtain aggregation kinetic data and examine the effect of MB in a reasonable timescale and with minimal stochasticity.

## SOD1 aggregate preparation

wtSOD1 aggregates were prepared by diluting 200 μM protein in 10 mM potassium phosphate buffer containing 0.5 M GuHCl and 5 mM DTT. Solutions were incubated at 60 °C under shaking conditions at 600 rpm (Ditabis MHR 23 shaker, Fisher Scientific) for 72 h. Mixtures containing MB and/or preformed seed were prepared by adding 400 μM to the buffer and/or 5% pre-formed SOD1 fibrils respectively.

## ThT fluorescence assays

Aggregation experiments were carried out with 200 μM apoSOD1 monomer in 10 mM potassium phosphate, 0.5 M GuHCl, 5 mM DTT buffer, 50 μM of Thioflavin T and 0–400 μM of MB. For seeded aggregation experiments, 5% of ultrasonicated fibrils were added into reaction mixtures and mixed thoroughly. Aliquots of 100 μl from aggregation mixtures were poured into a 96-well low-binding microplate (Corning 3881). ThT fluorescence (excitation at 442 nm, emission at 482 nm) was monitored at 60 °C in continuous shaking conditions every 5 or 10 min in Biotek Synergy H4 Multi-Mode microplate reader. Seeded aggregation experiments were carried out at 60 °C under quiescent conditions in RT-PCR analyzer (Rotor Gene Q).

## Dye binding assays

Before any dye-binding assay, 2 ml of SOD1 aggregates were dialyzed 4 times in 1 L of 10 mM potassium phosphate buffer, pH 7.4 at room temperature. In Congo Red binding assay, 10 ul of aggregates are mixed with 1 ml 10 mM potassium phosphate buffer containing
10 µM Congo Red. The sample is incubated for 15 min at room temperature, then light absorption in the 400–700 nm range is measured (Shimadzu UV-1800 spectrophotometer). Nile Red binding assay is performed the same way, except that 10 µM of Nile Red is added to the buffer instead of Congo Red. The dye is excited with 530 nm light and fluorescence emission is measured at 550–800 nm range. In ThT assay, 50 µM of ThT is added instead of Congo Red; fluorescence emission is detected in 450–600 nm range after excitation with 440 nm light beam.

## Atomic force microscopy

30 µl of the sample was deposited on freshly cleaved mica and left to adsorb for 5 min; followed by rinsing with distilled water and gentle drying using airflow. AFM images were recorded in the Tapping-in-Air mode at a drive frequency of 300 kHz, with the Dimension Icon microscope (Bruker).

## Fourier transform infrared spectroscopy

A buffer of prepared fibers was exchanged into $D_2O$ by a series of centrifugation and rinsing steps. Shortly, fibrils were pelleted in a microcentrifuge tube by spinning down at 12 000 rpm for 20 mins. The supernatant was removed and one mL of $D_2O$ was poured on the pellet. The rinsing cycle was repeated 4–5 times, followed by resuspension of fibrils in 300 µl $D_2O$ during the last step. Fibrils were then sonicated for 1 min at 20% MS72 probe power (Bandelin). FTIR spectra were recorded in a vacuum FTIR spectrophotometer (Bruker). Obtained spectra were processed by subtracting $D_2O$ reference spectrum and then normalized using GRAMS/AI software (Fisher Scientific).

## Data normalization

Experimental data were normalized using (1) equation:

$$I_{norm} = \frac{I - I_a}{I_b - I_a} 100 \tag{1}$$

where $I_{norm}$—normalized ThT fluorescence, I—fluorescence intensity at a certain time point, $I_a$ and $I_b$—minimum and maximum of fluorescence intensity.

## RESULTS

### Spontaneous SOD1 aggregation

The impact of MB was first investigated on the spontaneous aggregation kinetics of SOD1, using a ThT fluorescence assay (Fig. 1A). According to the data, SOD1 aggregation curves have a typical sigmoidal shape and in the control sample (Fig. 1A) the endpoint ThT fluorescence emission value is significantly higher than the background fluorescence, suggesting that long intermolecular β-sheet structures are being formed. The evidence of fibril formation is shown in an AFM image (Fig. 1B), where several µm-long fibrillar structures are visible. MB shows an inhibitory effect on SOD1 aggregation in a concentration-dependent manner, affecting the nucleation phase (as seen by the increase in lag time). Its effect on the quantity of fibrils formed cannot be determined accurately, as MB is known to partially quench the fluorescence of ThT
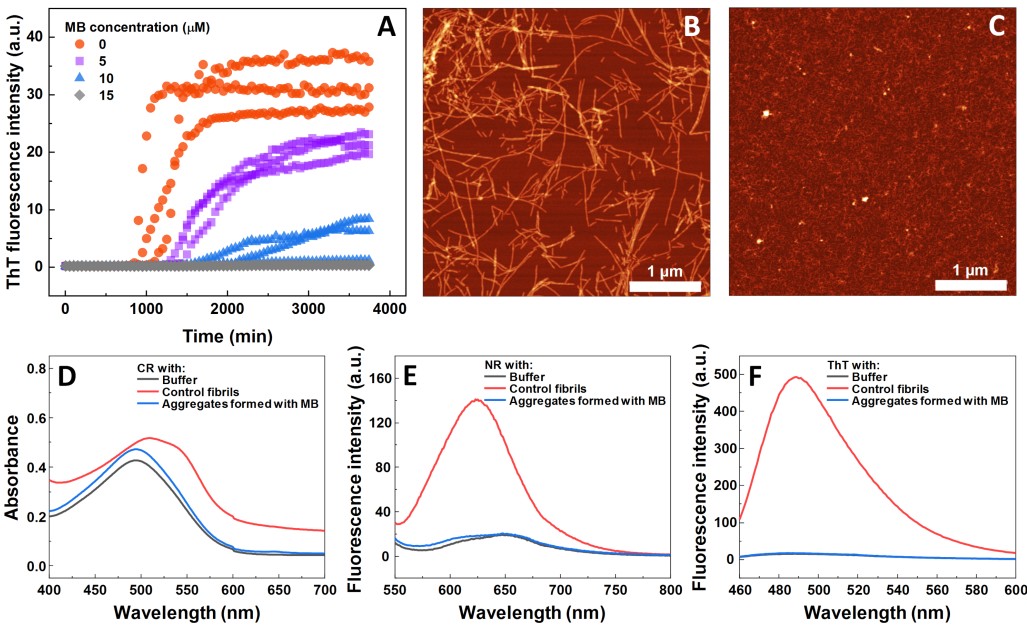

**Figure 1  Spontaneous SOD1 aggregation.** SOD1 aggregation kinetics followed by ThT (A); SOD1 aggregated in 10 mM potassium phosphate buffer with 0.5 M GuHCl and 5 mM DTT with 0–15 μM MB, pH 7.4. AFM images of aggregates formed without (B) and with (C) 15 μM MB. CR absorbance spectrum (D), as well as NR (E) and ThT (F) fluorescence spectra of aggregates formed with MB.

(*Ziaunys, Mikalauskaite & Smirnovas, 2019*). Because of the possibility of MB interacting with ThT and affecting both the total fluorescence, as well as aggregation kinetic monitoring, the aggregation reaction was simultaneously tracked by measuring sample absorbance and ThT fluorescence (Fig. S3). Absorbance measurements had a good correlation with the ThT fluorescence assay. Due to this quenching effect, AFM was also employed to confirm aggregate formation and morphology. As the concentration of methylene blue reaches 15 μM, SOD1 is no longer capable to aggregate into amyloid fibrils within the timescale of the experiment (Fig. 1A and Fig. S3). Interestingly, in high enough concentrations of MB, SOD1 no longer aggregates into amyloid fibrils, but rather forms small oligomers/protofibrils, as seen from an AFM image (Fig. 1C). Furthermore, these small aggregates are not capable of binding several widely used dyes for amyloid detection (Figs. 1D–1F). In the presence of these aggregates, the absorbance spectra of Congo Red dye contains a single peak at 490 nm and the spectrum is nearly identical to the spectrum of Congo red in just the buffer solution, indicating the presence of an unbound dye, whereas upon fibril binding, a typical peak at 540 nm appears (Fig. 1D). Nile Red and ThT assays also do not demonstrate an enhanced fluorescence intensity when aggregates are formed in the presence of MB (Figs. 1E, 1F).

## Seeded SOD1 aggregation

Seeded aggregation experiments were conducted to find out whether MB is capable of modifying the SOD1 aggregation pathway when pre-formed fibrils (seeds) are present. In

the presence of seeds, SOD1 aggregates immediately without a lag time, (Figs. 2A–2F). At MB concentrations up to 50 µM, the process of aggregation reaches a plateau within 150 min (Figs. 2A–2C). At higher MB concentrations, the initial increase is followed by a drop in a ThT fluorescence emission signal (Figs. 2D–2F). The morphology of aggregates formed this way is shown in Fig. 2H. The most abundant species are small, round-shaped aggregates similar to those formed during the process of spontaneous aggregation (shown in Fig. 1C) and look significantly different from fibrils formed in a control sample (Fig. 2G). Several fibers, visible in Fig. 2H, could be traces of the primary fibrillar seed, used to initiate the elongation process. These aggregates, obtained from seeded fibrillization experiments, are also incapable of binding amyloid specific dyes Congo Red, Nile Red and ThT (Fig. 3). Unlike in the spontaneous aggregation experiment, where fibrillization was completely stopped at 15 µM MB, seeds were still capable of elongating at MB concentrations above 15 µM.

## Seeding properties of non-fibrillar SOD1 aggregates formed with MB

To find out what are the seeding properties of aggregates formed in the presence of 400 µM MB, the following aggregation kinetics experiments were carried out. Various fractions (1–10%) of the aforementioned aggregates were used as seeds to initiate SOD1 monomer fibrillization (Fig. 4A). In particular, round-shaped aggregates, formed by incubating 200 µM SOD1 with 400 µM MB (shown in Fig. 1C), were mixed with monomeric SOD1 in the aggregation favoring buffer and fibrillization kinetics were followed by a ThT assay. As seen in Fig. 4A, ThT fluorescence remained at the baseline level even in the presence of 10% seed and no fibrils were formed (Fig. 4C), whereas in the control samples (SOD1 aggregation initiated by SOD1 fibrils) an immediate rise in ThT signal occurs and fibrils are observed by AFM (Fig. 4B).

## MB effect on pre-formed SOD1 fibrils

In order to test the hypothesis that MB is capable of disassembling SOD1 fibrils, fibers were incubated with MB in a ratio of 1:2 under optimized aggregation conditions. According to AFM data presented in Fig. 5A, SOD1 fibers are present even after 3 days of incubation, indicating that once aggregation occurs, MB does not disassemble fibrils within the experimental timescale.

To determine alterations in the secondary structure of SOD1 fibrils upon incubation with MB, FTIR spectra of the aggregates were recorded (Figs. 5B, 5C). SOD1 fibers have a strong signal at 1,624 cm$^{-1}$ (with the main minimum of the second derivative at 1,622 cm$^{-1}$ and a weaker one at 1,638 cm$^{-1}$), indicating the dominance of an intermolecular β-sheet structure. A second derivative minimum at 1,669 cm$^{-1}$, corresponding to a polypeptide chain turn motif is also observed. On the contrary, in the sample of non-aggregated SOD1, a clear single peak at 1,633 cm$^{-1}$ (with the main minimum of the second derivative at 1628 cm$^{-1}$ and a weaker one at 1,646 cm$^{-1}$), is observed, indicating the dominance of a weaker intramolecular β-sheet structure (*Barth, 2007*). In the case of preformed fibrils that were incubated with MB for 3 days, the FTIR spectrum main maximum is at 1,632 cm$^{-1}$, while the second derivative major minimum is at 1,624 cm$^{-1}$, with a weaker one at 1,634 cm$^{-1}$.

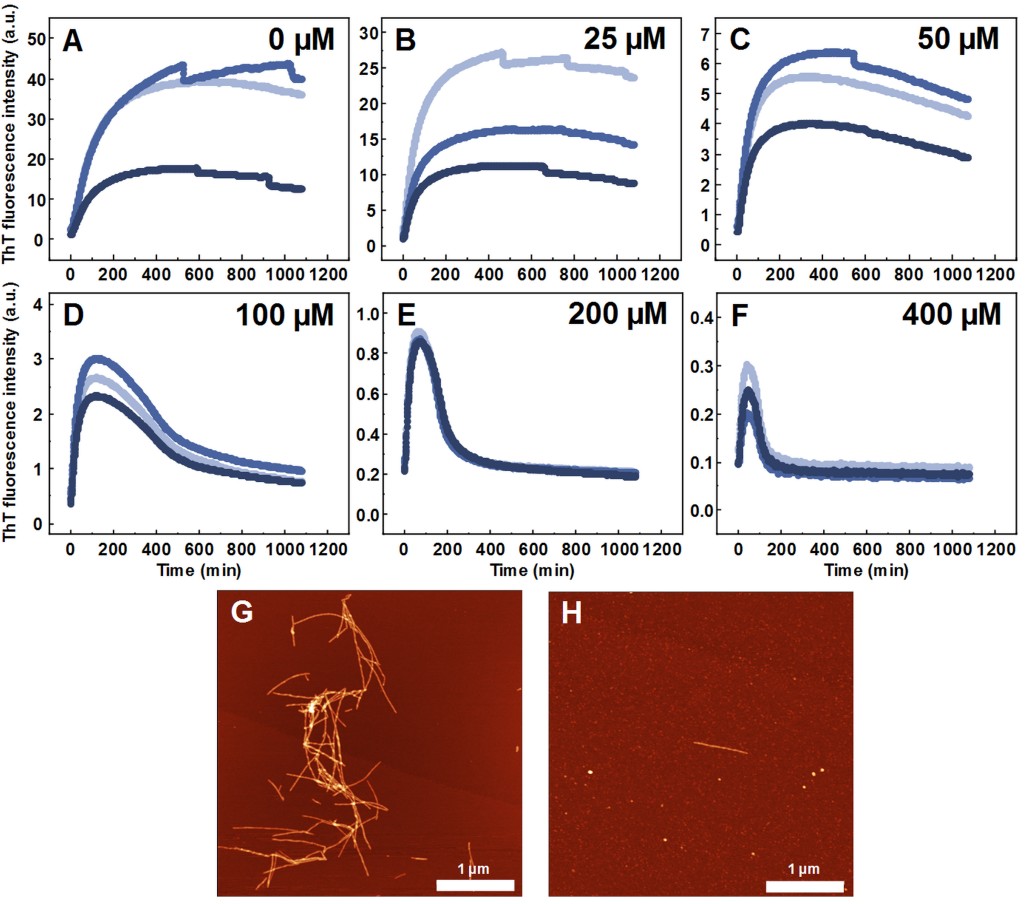

**Figure 2** **Seeded SOD1 aggregation.** Aggregation kinetics followed by ThT, when 0 (A), 25 (B), 50 (C), 100 (D), 200 (E) and 400 µM (F) of MB is present in the sample; SOD1 aggregated with 5% (of total protein in the sample) seed in 10 mM potassium phosphate buffer with 0.5 M GuHCl and 5 mM DTT with 0–400 µM methylene blue, pH 7.4. Three repeats for each condition are represented by different colors. AFM images of aggregates formed without (G) and with (H) 400 µM methylene blue.

This implies that the MB-treated fibrils have a secondary structure that is in between the non-aggregated protein and the control fibrils. Interestingly, fibrils incubated with MB were also incapable of binding Congo Red, Nile Red and ThT (Figs. 5D–5F), similarly to previously described non-fibrilllar aggregates.

As MB affects SOD1 fiber dye-binding properties, which may be related to its interaction with the fibril surface, it is possible that MB-incubated fibrils possess altered seeding properties as well. Therefore, this hypothesis was tested by following seeded aggregation kinetics with a ThT fluorescence assay. According to the data (Figs. 6A–6C), MB-incubated SOD1 fibrils elongate at a similar rate as control fibers. These results show that upon incubation with MB, fibril ends remain as active aggregation centers and initiate self-replication of their structure. These ThT-positive aggregates are in the form of elongated fibers (Fig. 6E) and are morphologically similar to fibers formed in a control aggregation mixture (Fig. 6D).

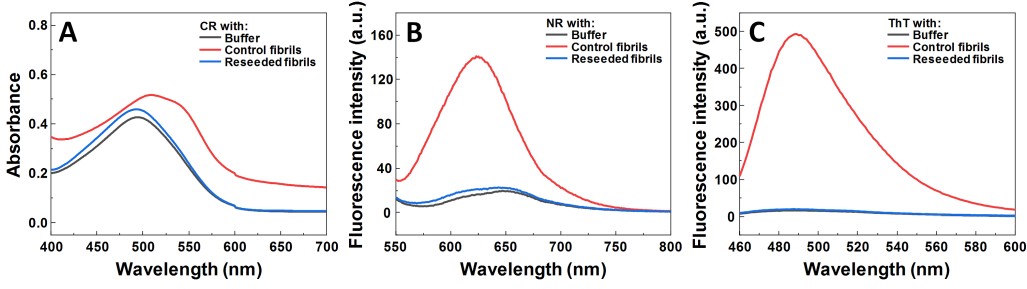

**Figure 3** **Dye binding assay of aggregates formed by reseeding SOD1 fibrils in the presence of MB.** CR (A) absorbance spectrum, as well as NR (B) and ThT (C) fluorescence spectra.

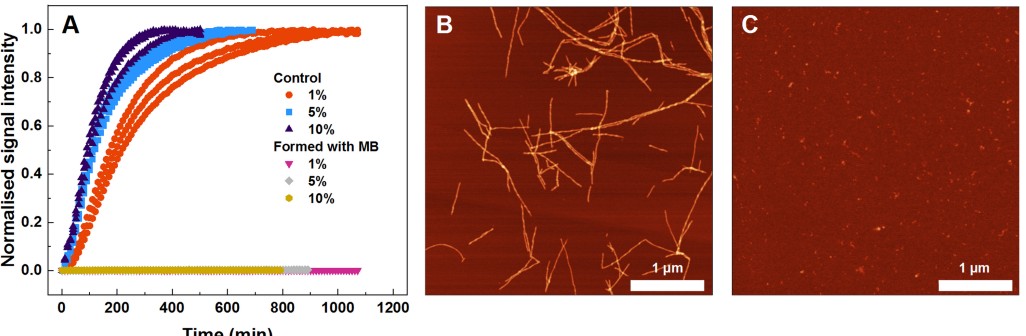

**Figure 4** **Seeded SOD1 aggregation, where aggregates formed with MB were used as seed.** Aggregation kinetics followed by ThT (A); SOD1 aggregated with 1–10% seed in 10 mM potassium phosphate buffer with 0.5 M GuHCl and 5 mM DTT, pH 7.4. The seed was prepared by incubating SOD1 with 400 µM methylene blue. Control samples contained an equivalent concentration of MB as the samples with the MB-formed aggregates. AFM images from the control experiment (B) and when fibrils formed with MB were used as seed (C).

## DISCUSSION

When examining the effect of MB on the spontaneous aggregation of SOD1, the data from fibrillization kinetics (Fig. 1A), AFM (Figs. 1B, 1C) and amyloid dye-binding assays (Figs. 1D–1F) suggest that MB not only affects the rate of SOD1 amyloid formation, but also has an impact on the final aggregate structure. The slower association rates, as well as the formation of round-shaped structures that can either be "off-pathway" oligomers or kinetically trapped pre-fibrillar aggregates imply that MB can act as a SOD1 aggregation pathway modulator. Furthermore, the aggregates formed with MB were used to test their seeding propensity (Fig. 4). The baseline level of ThT fluorescence intensity indicates the absence of β-sheet structures and leads to the conclusion that these structures are incapable of incorporating SOD1 monomers in a templated fashion and therefore are not amyloid-like.

As MB was capable of inhibiting SOD1 spontaneous fibrillization, its effect was also tested in seeded aggregation conditions (Fig. 2). In this case, two types of effects were
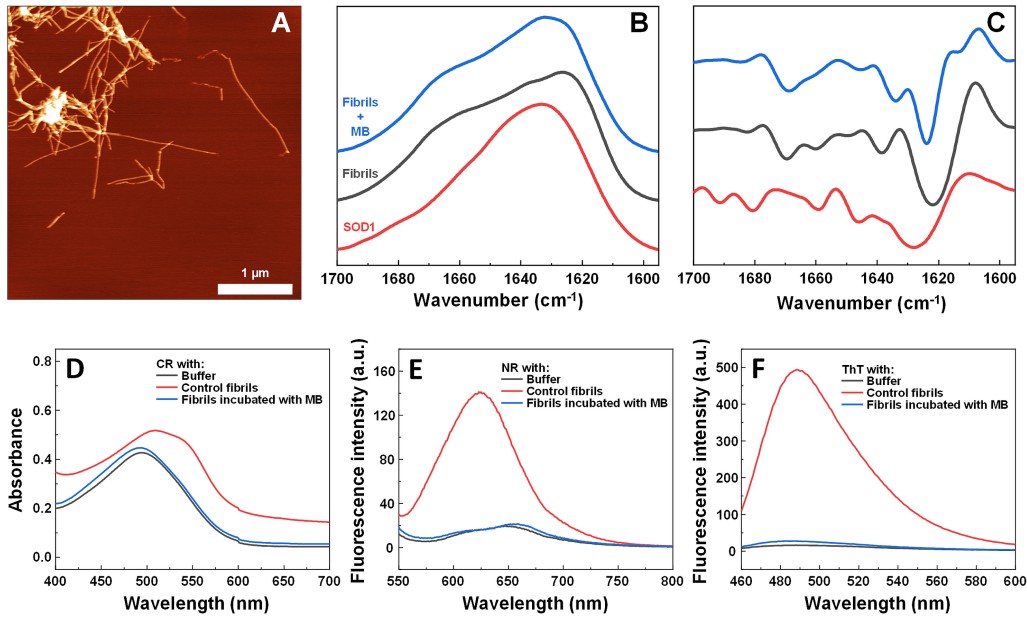

**Figure 5    Effect of MB on SOD1 fibril morphology and FTIR spectra.** AFM image of 200 μM preformed SOD1 fibrils incubated with 400 μM methylene blue for 3 days (A), FTIR (B) and second derivative (C) spectra of SOD1 fibrils and monomer. CR absorbance spectrum (D), as well as NR (E) and ThT (F) fluorescence spectra of aggregates formed with MB.

observed. Firstly, 15 μM of MB no longer had the capacity to fully inhibit fibril growth and much higher concentrations were required to make an effect on SOD1 aggregation kinetics (Figs. 2A–2F). This suggests that while MB is capable of modulating nucleation and fibril elongation, it is more potent at inhibiting the former process, rather than the latter one. Secondly, the initial increase in ThT fluorescence intensity was followed by a sudden drop. This could be the result of multiple possible events. The initial hypothesis was that a drop in ThT signal occurs due to a decrease in the fibril mass. However, fibrils incubated with MB demonstrate similar seeding capability as non-incubated counterparts, as indicated by aggregation kinetics experiments (Figs. 6A–6C). This is further supported by AFM studies (Fig. 6E), showing the dominance of fibrillar material after pre-formed fibril treatment with MB. According to the second hypothesis, ThT and MB interact with each other either in solution or on the surface of SOD1 aggregates, as a similar phenomenon was shown to occur on the surface of insulin fibrils (*Ziaunys, Mikalauskaite & Smirnovas, 2019*). Such interaction may result in ThT fluorescence quenching. However, this scenario does not explain why the fluorescence intensity drop occurs only after a certain time period in a MB-concentration dependent manner (Figs. 2A–2F). An alternative explanation would be that MB requires a certain amount of time to change the surface or structure of the formed aggregates, thus impeding their ability to bind amyloid-specific dyes. If this process is significantly slower than fibril elongation, then the drop in intensity would appear after some signal growth is first observed, which is the case in this seeded aggregation

Musteikyte et al. (2020), *PeerJ*, DOI 10.7717/peerj.9719
10/17

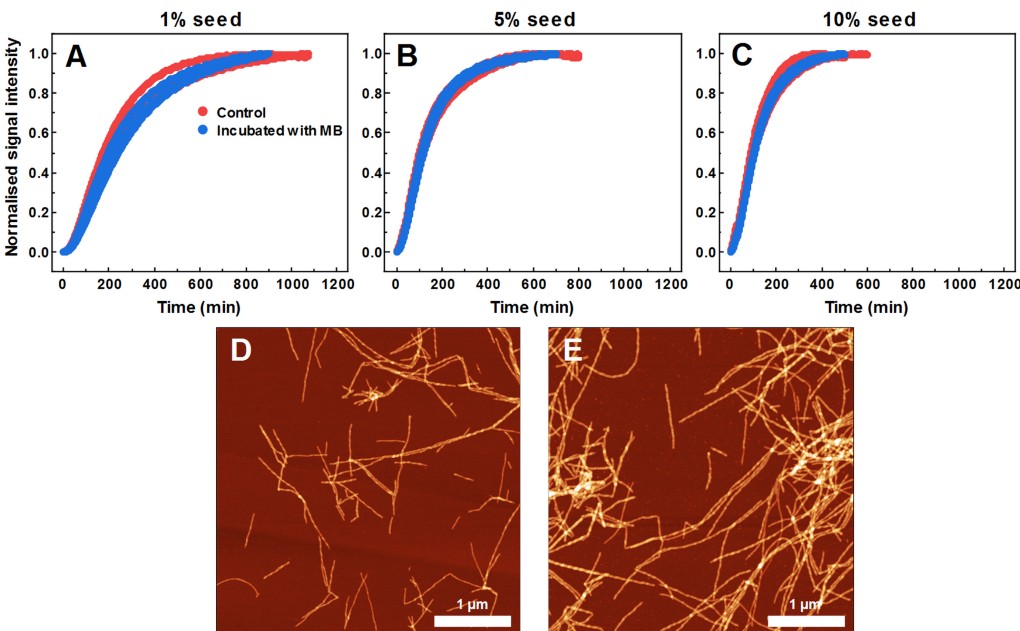

**Figure 6** **Seeded SOD1 aggregation, when fibrils incubated with MB were used as seed.** Aggregation kinetics followed by ThT; SOD1 aggregated with 1% (A), 5% (B) and 10% (C) seed in 10 mM potassium phosphate buffer with 0.5 M GuHCl and 5 mM DTT, pH 7.4. Control samples contained an equivalent concentration of MB as the samples with MB-treated fibrils. AFM images of aggregates formed with control (D) and MB-treated fibrils (E).

experiment. Because of this effect, the kinetic data may not be an accurate representation of the aggregation processes at high MB concentrations.

The hypothesis that MB can change the surface or structure of preformed fibrils was examined by incubating SOD1 fibrils with MB and then comparing their FTIR spectra (Figs. 5B, 5C), morphology (Fig. 5A), seeding propensity (Figs. 6A–6C), as well as dye-binding ability (Figs. 5D–5F). According to the FTIR spectra, at a first glance it seems like the secondary structure of MB-incubated fibrils lies in between the fibrils and monomeric species, as the main maximum in the FTIR spectrum is at 1,632 cm$^{-1}$. However, a second derivative profile of SOD1 MB-treated fiber spectrum (Fig. 5C) is similar to the spectrum of untreated fibrils—the major minimum at 1,624 cm$^{-1}$ and a weaker one at 1,634 cm$^{-1}$ suggest the presence of intermolecular β-sheets, whereas a minimum at 1669 cm$^{-1}$ indicates the presence of polypeptide chain turns. This shows that MB slightly alters SOD1 fibril FTIR spectra, but does not refold their secondary structure. Data from seeding kinetics (Figs 6A–6C) and AFM images (Figs. 6D, 6E) are in agreement with this conclusion, as their morphology and seeding propensity remain similar to fibrils from the control sample. The major difference is that MB-incubated fibers are incapable of binding amyloid-specific dyes (Figs. 5D–5F), similarly to aggregates formed with MB during spontaneous aggregation. Taken together, this leads to the conclusion that once SOD1 fibrils are formed, MB is only capable of interacting with their surface, thus slightly altering their FTIR spectra and dye-binding ability, while having no effect on their morphology and seeding propensity.

## CONCLUSIONS

MB has the capacity to not only affect spontaneous SOD1 amyloid aggregation, leading to off-pathway species, but it also impedes preformed fibril elongation. Together with the previously reported inhibitory effect of MB on aggregation of a series of amyloid-forming proteins, our data supports MB as a potential compound in the search for anti-amyloid drugs.

## ACKNOWLEDGEMENTS

The authors acknowledge Prof. G. Niaura from the Center of Physical Sciences and Technology for access to FTIR.

### Funding

This work was supported by the European Social Fund (project No 09.3.3.-LMT-K-712-03-0059) under grant agreement with the Research Council of Lithuania. The funders had no role in study design, data collection and analysis, decision to publish, or preparation of the manuscript.

### Grant Disclosures

The following grant information was disclosed by the authors:
European Social Fund (project No 09.3.3.-LMT-K-712-03-0059) under grant agreement with the Research Council of Lithuania.

### Competing Interests

The authors declare there are no competing interests.

### Author Contributions

- Greta Musteikyte conceived and designed the experiments, performed the experiments, analyzed the data, prepared figures and/or tables, authored or reviewed drafts of the paper, and approved the final draft.
- Mantas Ziaunys analyzed the data, prepared figures and/or tables, authored or reviewed drafts of the paper, and approved the final draft.
- Vytautas Smirnovas conceived and designed the experiments, analyzed the data, authored or reviewed drafts of the paper, and approved the final draft.

### Data Availability

Kinetic, dye binding, and FTIR data and AFM raw images are available as Supplemental Files.

### Supplemental Information

Supplemental information for this article can be found online at http://dx.doi.org/10.7717/peerj.9719#supplemental-information.

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
