# Peer review of "Methylene blue inhibits nucleation and elongation of SOD1 amyloid fibrils"

_PeerJ, doi:10.7717/peerj.9719_

## Round 0.1 · original submission · Major Revisions

Please address issues raised by both reviewers and revise your manuscript accordingly. Please note that careful attention should be paid to the comments from reviewer #2.

Reviewer 1 ·

Basic reporting

The article reporting the inhibitory action of Methylene Blue on nucleation and elongation of SOD1 amyloid fibrils. However, there are major problems in this manuscript which are as follows:
1. There is recent work on effect of this same dye on fibrillation in lysozyme which has not been cited.
2. The results of Congo Red, Nile Red Red assay should be discussed in detail in the main manuscript and not in the Supplementary section.

Experimental design

There are some major flaws in the experimental design which need to be addressed. e.g.
ANS assay should be performed in fluorescence.
The CD experiments should be performed to ascertain the conformational changes and the CD curves should be deconvoluted to decipher the changes in the secondary structure of the protein.
Intrinsic fluorescence experiments should also be performed.
The interaction of the dye with the protein should have been done at first using UV-Vis, steady-state, and time-dependent fluorescence as well as Isothermal titration calorimetry.
Molecular docking should also be performed.
Above all, the effect of methylene blue should be investigated on proper neurological cell lines if it is to have any potential anti-amyloidogenic activity.

Validity of the findings

In the absence of the experiments suggested above the paper appears to be flawed fundamentally and the results reported are too thin and premature to jump to any conclusions.

Additional comments

The work reported is not sufficient to validate the anti-amyloidogenic property of the dye and the authors should attempt to study the effect of methylene blue on a more realistic model preferably some apt cell line or animal model before they jump to any conclusion regarding the potential inhibitory effect of the dye on amyloid fibrillation.

Reviewer 2 ·

Basic reporting

In the present manuscript, Musteikyte and co-workers perform an in vitro study on the activity of the dye Methylene blue as an inhibitor of SOD1 amyloid formation under denaturing conditions. The text is clear; the English is mostly correct. The introduction and associated literature are adequate. Raw data is shared, and the Figures in the main text and supplementary files are of good quality.

Experimental design

There are two main problems with the methodology in the present manuscript.

1)The aggregation conditions are far from physiological conditions and question the nativeness of the protein at the beginning of the reaction. First, the protein is incubated in 5 mM DTT, a very high concentration of reducing agent that should reduce the intrasubunit disulfide. Besides, the protein is demetallated, which also destabilizes it. On top of that, they add 0,5 GuHCl, and the protein solutions are incubated at 60 oC with intense agitation. The authors rationalize that it is necessary for a reproducible on-pathway aggregation. Still, I am afraid that under these extreme conditions, the protein is aggregating from a significantly unfolded state, questioning the relevance of the acquired data in a biological context and the applicability of MB as a drug for SOD1 related diseases. The authors should demonstrate that, under the assay conditions, the protein is initially significantly folded and, ideally, active.

2) The aggregation kinetics are followed with Th-T, whereas it is known that MB interferes significantly with the amyloid dye's signal. This is also the case for CR; MB has been reported to compete for the binding of Nile Red to fibrils and aggregates. This lets us only with the AFM data, which are convincing but should be complemented with at least an alternative biophysical technique. Orthogonal light scattering kinetics or DLS measurements at selected time points might help.

Validity of the findings

Apart from the methodological aspects mentioned above, which I think are important:

1-It is said that MB acts at the nucleation stage, but if I am not wrong, this is deduced from the direct observation of the curves. Calculation of the respective nucleation and elongation rate constants would provide support to this assumption.

2- In the seeding assays, the amount of MB equal or even is several-fold higher than this of Th-T, raising question on whether the quenching effect mask any effect of the dye on seeded aggregation. This might explain the strange curves obtained at concentrations higher than 100 uM of MB. One of the explanations the authors provide to this phenomenon is: "An alternative explanation would be that MB changes the surface or structure of the formed aggregates, thus impeding their ability to bind amyloid-specific dyes." If this is the case, this phenomenon would have important implications for the kinetic assays recorded in the manuscript.


3- In the "Seeding properties of non-fibrillar SOD1 aggregates formed with MB" section, as long I understand, there is not any step to purify the aggregates formed in the presence of 400 uM of MB. If this is the case and the solution is directly used as a seed in the lower 1% proportion, the seeded reaction would have 4 uM of MB and in the 10% seeded reaction 40 uM, this makes difficult to elucidate if the dye contributes at least in part to abrogate the seeding. In any case, controls should be included.

4- I am not sure that the differences in the FTIR spectra of fibrils formed in the presence and absence of MB are enough to derive the structural conclusions the authors extract. The deconvolution of the absorbance spectra on their components might help to support such findings.

Overall I am not convinced that from the experiments in the present form of the manuscript, it can be concluded that MB might be a therapeutic lead for the treatment of SOD1 associated amyloid disorders.

---

## Round 0.2 · accepted · Accept

All critiques were adequately addressed and the manuscript was revised accordingly. Therefore, amended version can be accepted now.